# How to Robustify Black-Box ML Models? A Zeroth-Order Optimization Perspective

**Yimeng Zhang**
Michigan State University

**Yuguang Yao**
Michigan State University

**Jinghan Jia**
Michigan State University

**Jinfeng Yi**
JD AI Research

**Mingyi Hong**
University of Minnesota

**Shiyu Chang**
UC Santa Barbara

**Sijia Liu**
Michigan State University

## Abstract

The lack of adversarial robustness has been recognized as an important issue for state-of-the-art machine learning (ML) models, e.g., deep neural networks (DNNs). Thereby, robustifying ML models against adversarial attacks is now a major focus of research. However, nearly all existing defense methods, particularly for robust training, made the *white-box* assumption that the defender has the access to the details of an ML model (or its surrogate alternatives if available), e.g., its architectures and parameters. Beyond existing works, in this paper we aim to address the problem of *black-box defense*: How to robustify a black-box model using just input queries and output feedback? Such a problem arises in practical scenarios, where the owner of the predictive model is reluctant to share model information in order to preserve privacy. To this end, we propose a general notion of defensive operation that can be applied to black-box models, and design it through the lens of denoised smoothing (DS), a first-order (FO) certified defense technique. To allow the design of merely using model queries, we further integrate DS with the zeroth-order (gradient-free) optimization. However, a direct implementation of zeroth-order (ZO) optimization suffers a high variance of gradient estimates, and thus leads to ineffective defense. To tackle this problem, we next propose to prepend an autoencoder (AE) to a given (black-box) model so that DS can be trained using variance-reduced ZO optimization. We term the eventual defense as ZO-AE-DS. In practice, we empirically show that ZO-AE-DS can achieve improved accuracy, certified robustness, and query complexity over existing baselines. And the effectiveness of our approach is justified under both image classification and image reconstruction tasks. Codes are available at `https://github.com/damon-demon/Black-Box-Defense`.

## 1 Introduction

ML models, DNNs in particular, have achieved remarkable success owing to their superior predictive performance. However, they often lack robustness. For example, imperceptible but carefully-crafted input perturbations can fool the decision of a well-trained ML model. These input perturbations refer to *adversarial perturbations*, and the adversarially perturbed (test-time) examples are known as *adversarial examples* or *adversarial attacks* (Goodfellow et al., 2015; Carlini & Wagner, 2017; Papernot et al., 2016). Existing studies have shown that it is not difficult to generate adversarial attacks. Numerous attack generation methods have been designed and successfully applied to *(i)* different use cases from the digital world to the physical world, *e.g.*, image classification (Brown et al., 2017; Li et al., 2019; Xu et al., 2019; Yuan et al., 2021), object detection/tracking (Eykholt et al., 2017; Xu et al., 2020; Sun et al., 2020), and image reconstruction (Antun et al., 2020; Raj et al., 2020; Vasiljević et al., 2021), and *(ii)* different types of victim models, *e.g.*, white-box models whose details can be accessed by adversaries (Madry et al., 2018; Carlini & Wagner, 2017; Tramer et al., 2020; Croce & Hein, 2020; Wang et al., 2021), and black-box models whose information is not disclosed to adversaries (Papernot et al., 2017; Tu et al., 2019; Ilyas et al., 2018a; Liang et al., 2021).

Given the prevalence of adversarial attacks, methods to robustify ML models are now a major focus in research. For example, adversarial training (AT) (Madry et al., 2018), which has been poised one of the most effective defense methods (Athalye et al., 2018), employed min-max optimization to minimize the worst-case (maximum) training loss induced by adversarial attacks. Extended from AT, various empirical defense methods were proposed, ranging from supervised learning, semi-supervised learning, to unsupervised learning (Madry et al., 2018; Zhang et al., 2019b; Shafahi et al., 2019; Zhang et al., 2019a; Carmon et al., 2019; Chen et al., 2020; Zhang et al., 2021). In addition to empirical defense, certified defense is another research focus, which aims to train provably robust ML models and provide certificates of robustness (Wong & Kolter, 2017; Raghunathan et al., 2018; Katz et al., 2017; Salman et al., 2019; 2020; 2021). Although exciting progress has been made in adversarial defense, nearly all existing works ask a defender to perform over *white-box* ML models (assuming non-confidential model architectures and parameters). However, the white-box assumption may restrict the defense application in practice. For example, a model owner may refuse to share the model details, since disclosing model information could hamper the owner's privacy, *e.g.*, model inversion attacks lead to training data leakage (Fredrikson et al., 2015). Besides the privacy consideration, the white-box defense built upon the (end-to-end) robust training (*e.g.*, AT) is computationally intensive, and thus is difficult to scale when robustifying multiple models. For example, in the medical domain, there exist massive pre-trained ML models for different diseases using hundreds of neuroimaging datasets (Sisodiya et al., 2020). Thus, robustly retraining all models becomes impractical. Taking the model privacy and the defense efficiency into consideration, we ask:

*Is it possible to design an adversarial defense over **black-box** models using only model queries?*

Extending adversarial defense to the black-box regime (that we call 'black-box defense') is highly non-trivial due to the challenge of black-box optimization (*i.e.*, learning over black-box models). To tackle this problem, the prior work (Salman et al., 2020) leveraged *surrogate models* as approximations of the black-box models, over which defense can be conducted following the white-

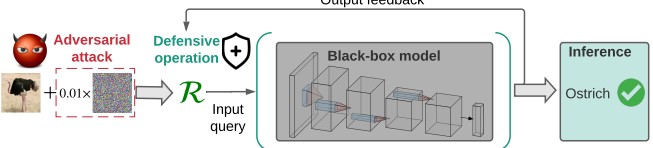

Figure 1: Illustration of defense against adversarial attacks for entirely black-box models.

box setup. Yet, this still requires to have access to the information on the victim model type and its function. In practice, those conditions could be difficult to achieve. For example, if the domain knowledge related to medicine or healthcare is lacking (Qayyum et al., 2020; Finlayson et al., 2019), then it will be difficult to determine a proper surrogate model of a medical ML system. Even if a black-box model estimate can be obtained using the model inversion technique (Kumar & Levine, 2019), a significantly large number of model queries are needed even just for tackling a MNIST-level prediction task (Oh et al., 2019). Different from (Salman et al., 2020), we study an *authentic black-box scenario*, in which the interaction between defender and model is only based on input-output function queries (see Fig. 1). To our best knowledge, this is the first work to tackle the problem of query-based black-box defense.

**Contributions.** We summarize our contributions below.

① (Formulation-wise) We formulate the problem of black-box defense and investigate it through the lens of zeroth-order (ZO) optimization. Different from existing works, our paper aims to design the restriction-least black-box defense and our formulation is built upon a query-based black-box setting, which avoids the use of surrogate models.

② (Methodology-wise) We propose a novel black-box defense approach, ZO AutoEncoder-based Denoised Smoothing (ZO-AE-DS), which is able to tackle the challenge of ZO optimization in high dimensions and convert a pre-trained non-robust ML model into a certifiably robust model using only function queries.

③ (Experiment-wise) We verify the efficacy of our method through an extensive experimental study. In the task of image classification, the proposed ZO-AE-DS significantly outperforms the ZO baseline built upon (Salman et al., 2020). For instance, we can improve the certified robust accuracy of ResNet-110 on CIFAR-10 from $19.16\%$ (using baseline) to $54.87\%$ (using ZO-AE-DS) under adversarial perturbations with $\ell_2$ norm less than $64/255$. We also empirically show that our proposal stays effective even in the task of image reconstruction.

## 2 RELATED WORK

**Empirical defense.** An immense number of defense methods have been proposed, aiming to improve model robustness against adversarial attacks. Examples include detecting adversarial attacks (Guo et al., 2017; Meng & Chen, 2017; Gong et al., 2017; Grosse et al., 2017; Metzen et al., 2017) and training robust ML models (Madry et al., 2018; Zhang et al., 2019b; Shafahi et al., 2019; Wong et al., 2020; Zhang et al., 2019a; Athalye et al., 2018; Cheng et al., 2017; Wong & Kolter, 2017; Salman et al., 2019; Raghunathan et al., 2018; Katz et al., 2017). In this paper, we focus on advancing the algorithm foundation of robust training over black-box models. Robust training can be broadly divided into two categories: empirical defense and certified defense. In the former category, the most representative method is AT (adversarial training) that formulates adversarial defense as a two-player game (between attacker and defender) (Madry et al., 2018). Spurred by AT, empirical defense has developed rapidly. For example, in (Zhang et al., 2019b), TRADES was proposed to seek the optimal trade-off between accuracy and robustness. In (Stanforth et al., 2019; Carmon et al., 2019), unlabeled data and self-training were shown effective to improve adversarial defense in both robustness and generalization. In (Shafahi et al., 2019; Wong et al., 2020; Zhang et al., 2019a; Andriushchenko & Flammarion, 2020), to improve the scalability of adversarial defense, computationally-light alternatives of AT were developed. Despite the effectiveness of empirical defense against adversarial attacks (Athalye et al., 2018), it lacks theoretical guarantee (known as 'certificate') for the achieved robustness. Thus, the problem of certified defense arises.

**Certified defense.** Certified defense seeks to provide a provably guarantee of ML models. One line of research focuses on post-hoc formal verification of a pre-trained ML model. The certified robustness is then given by a 'safe' input perturbation region, within which any perturbed inputs will not fool the given model (Katz et al., 2017; Ehlers, 2017; Bunel et al., 2018; Dutta et al., 2017). Since the exact verification is computationally intensive, a series of work (Raghunathan et al., 2018; Dvijotham et al., 2018; Wong & Kolter, 2017; Weng et al., 2018a;b; Wong et al., 2018) proposed 'incomplete' verification, which utilizes convex relaxation to over-approximate the output space of a predictive model when facing input perturbations. Such a relaxation leads to fast computation in the verification process but only proves a lower bound of the exact robustness guarantee. Besides the post-hoc model verification with respect to each input example, another line of research focuses on in-processing certification-aware training and prediction. For example, randomized smoothing (RS) transforms an empirical classifier into a provably robust one by convolving the former with an isotropic Gaussian distribution. It was shown in (Cohen et al., 2019) that RS can provide formal guarantees for adversarial robustness. Different types of RS-oriented provable defenses have been developed, such as adversarial smoothing (Salman et al., 2019), denoised smoothing (Salman et al., 2020), smoothed ViT (Salman et al., 2021), and feature smoothing (Addepalli et al., 2021).

**Zeroth-order (ZO) optimization for adversarial ML.** ZO optimization methods are gradient-free counterparts of first-order (FO) optimization methods (Liu et al., 2020b). They approximate the FO gradients through function value based gradient estimates. Thus, ZO optimization is quite useful to solve black-box problems when explicit expressions of their gradients are difficult to compute or infeasible to obtain. In the area of adversarial ML, ZO optimization has become a principled approach to generate adversarial examples from black-box victim ML models (Chen et al., 2017; Ilyas et al., 2018a;b; Tu et al., 2019; Liu et al., 2019; 2020a; Huang & Zhang, 2020; Cai et al., 2020; 2021). Such ZO optimization-based attack generation methods can be as effective as state-of-the-art white-box attacks, despite only having access to the inputs and outputs of the targeted model. For example, the work (Tu et al., 2019) leveraged the white-box decoder to map the generated low-dimension perturbations back to the original input dimension. Inspired by (Tu et al., 2019), we leverage the autoencoder architecture to tackle the high-dimension challenge of ZO optimization in black-box defense. Despite the widespread application of ZO optimization to black-box attack generation, few work studies the problem of black-box defense.

## 3 PROBLEM FORMULATION: BLACK-BOX DEFENSE

In this section, we formulate the problem of black-box defense, *i.e.*, robustifying black-box ML models without having any model information such as architectures and parameters.

**Problem statement.** Let $f_{\boldsymbol{\theta}_{\mathrm{bb}}}(\mathbf{x})$ denote a pre-defined _black-box (bb) predictive model_, which can map an input example $\mathbf{x}$ to a prediction. In our work, $f_{\boldsymbol{\theta}_{\mathrm{bb}}}$ can be either an image classifier or an image reconstructor. For simplicity of notation, we will drop the model parameters $\boldsymbol{\theta}_{\mathrm{bb}}$ when referring to a black-box model. The _threat model_ of our interest is given by norm-ball constrained adversarial attacks (Goodfellow et al., 2015). To defend against these attacks, existing approaches commonly require the white-box assumption of $f$ (Madry et al., 2018) or have access to white-box surrogate models of $f$ (Salman et al., 2020). Different from the prior works, we study the problem of _black-box defense_ when the owner of $f$ is not able to share the model details. Accordingly, the only mode of interaction with the black-box system is via submitting inputs and receiving the corresponding predicted outputs. The formal statement of black-box defense is given below:

> **(Black-box defense)** Given a black-box base model $f$, can we develop a defensive operation $\mathcal{R}$ using just input-output _function queries_ so as to produce the robustified model $\mathcal{R}(f)$ against adversarial attacks?

**Defensive operation.** We next provide a concrete formulation of the defensive operation $\mathcal{R}$. In the literature, two principled defensive operations were used: ($\mathcal{R}_1$) end-to-end AT (Madry et al., 2018; Zhang et al., 2019b; Cohen et al., 2019), and ($\mathcal{R}_2$) prepending a defensive component to a base model (Meng & Chen, 2017; Salman et al., 2020; Aldahdooh et al., 2021). The former ($\mathcal{R}_1$) has achieved the state-of-the-art robustness performance (Athalye et al., 2018; Croce & Hein, 2020) but is not applicable to black-box defense. By contrast, the latter ($\mathcal{R}_2$) is more compatible with black-box models. For example, _denoised smoothing_ (DS), a recently-developed $\mathcal{R}_2$-type approach (Salman et al., 2020), gives a certified defense by prepending a custom-trained denoiser to the targeted model. In this work, we choose DS as the backbone of our defensive operation (Fig. 2).

In DS, a denoiser is integrated with a base model $f$ so that the augmented system becomes resilient to Gaussian noise and thus plays a role similar to the RS-based certified defense (Cohen et al., 2019). That is, DS yields

$$\mathcal{R}(f(\mathbf{x})) := f(D_{\boldsymbol{\theta}}(\mathbf{x})), \qquad (1)$$

where $D_{\boldsymbol{\theta}}$ denotes the learnable denoiser (with

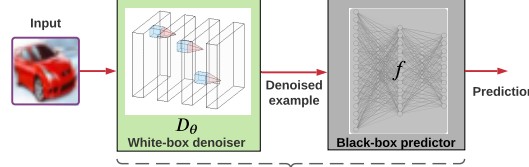

Figure 2: DS-based black-box defense.

parameters $\boldsymbol{\theta}$) prepended to the (black-box) predictor $f$. Once $D_{\boldsymbol{\theta}}$ is learned, then the DS-based smooth classifier, $\arg\max_c \mathbb{P}_{\boldsymbol{\delta}\in\mathcal{N}(\mathbf{0},\sigma^2\mathbf{I})}[\mathcal{R}(f(\mathbf{x}+\boldsymbol{\delta}))=c]$, can achieve certified robustness, where $c$ is a class label, $\boldsymbol{\delta}\in\mathcal{N}(\mathbf{0},\sigma^2\mathbf{I})$ denotes the standard Gaussian noise with variance $\sigma^2$, and $\arg\max_c \mathbb{P}_{\boldsymbol{\delta}\in\mathcal{N}(\mathbf{0},\sigma^2\mathbf{I})}[f(\mathbf{x}+\boldsymbol{\delta})=c]$ signifies a smooth version of $f$.

Based on (1), the goal of black-box defense becomes to find the optimal denoiser $D_{\boldsymbol{\theta}}$ so as to achieve satisfactory accuracy as well as adversarial robustness. In the FO learning paradigm, Salman et al. (2020) proposed a stability regularized denoising loss to train $D_{\boldsymbol{\theta}}$:

$$\underset{\boldsymbol{\theta}}{\text{minimize}} \quad \mathbb{E}_{\boldsymbol{\delta}\in\mathcal{N}(\mathbf{0},\sigma^2\mathbf{I}),\mathbf{x}\in\mathcal{U}} \underbrace{\|D_{\boldsymbol{\theta}}(\mathbf{x}+\boldsymbol{\delta})-\mathbf{x}\|_2^2}_{:=\ell_{\mathrm{Denoise}}(\boldsymbol{\theta})} + \gamma\mathbb{E}_{\boldsymbol{\delta},\mathbf{x}} \underbrace{\ell_{\mathrm{CE}}(\mathcal{R}(f(\mathbf{x}+\boldsymbol{\delta})),f(\mathbf{x}))}_{:=\ell_{\mathrm{Stab}}(\boldsymbol{\theta})}, \qquad (2)$$

where $\mathcal{U}$ denotes the training dataset, the first objective term $\ell_{\mathrm{Denoise}}(\boldsymbol{\theta})$ corresponds to the mean squared error (MSE) of image denoising, the second objective term $\ell_{\mathrm{Stab}}(\boldsymbol{\theta})$ measures the prediction stability through the cross-entropy (CE) between the outputs of the denoised input and the original input, and $\gamma > 0$ is a regularization parameter that strikes a balance between $\ell_{\mathrm{Denoise}}$ and $\ell_{\mathrm{Stab}}$.

We remark that problem (2) can be solved using the FO gradient descent method if the base model $f$ is fully disclosed to the defender. However, the black-box nature of $f$ makes the gradients of the stability loss $\ell_{\mathrm{Stab}}(\boldsymbol{\theta})$ infeasible to obtain. Thus, we will develop a _gradient-free_ DS-oriented defense.

## 4 METHOD: A SCALABLE ZEROTH-ORDER OPTIMIZATION SOLUTION

In this section, we begin by presenting a brief background on ZO optimization, and elaborate on the challenge of black-box defense in high dimensions. Next, we propose a novel ZO optimization-based DS method that can not only improve model query complexity but also lead to certified robustness.

**ZO optimization.** In ZO optimization, the FO gradient of a black-box function $\ell(\mathbf{w})$ (with a $d$-dimension variable $\mathbf{w}$) is approximated by the difference of two function values along a set of random direction vectors. This leads to the randomized gradient estimate (RGE) (Liu et al., 2020b):

$$\hat{\nabla}_{\mathbf{w}}\ell(\mathbf{w}) = \frac{1}{q}\sum_{i=1}^{q}\left[\frac{d}{\mu}\left(\ell(\mathbf{w}+\mu\mathbf{u}_i) - \ell(\mathbf{w})\right)\mathbf{u}_i\right], \tag{3}$$

where $\{\mathbf{u}_i\}_{i=1}^{q}$ are $q$ random vectors drawn independently and uniformly from the sphere of a unit ball, and $\mu > 0$ is a given small step size, known as the smoothing parameter. The rationale behind (3) is that it provides an *unbiased* estimate of the FO gradient of the Gaussian smoothing version of $\ell$ (Gao et al., 2018), with variance in the order of $O(\frac{d}{q})$ (Liu et al., 2020b). Thus, a large-scale problem (with large $d$) yields a large variance of RGE (3). To reduce the variance, a large number of querying directions (*i.e.*, $q$) is then needed, with the worst-case query complexity in the order of $O(d)$. If $q = d$, then the least estimation variance can be achieved by the coordinatewise gradient estimate (CGE) (Lian et al., 2016; Liu et al., 2018):

$$\hat{\nabla}_{\mathbf{w}}\ell(\mathbf{w}) = \sum_{i=1}^{d}\left[\frac{\ell(\mathbf{w}+\mu\mathbf{e}_i) - \ell(\mathbf{w})}{\mu}\mathbf{e}_i\right], \tag{4}$$

where $\mathbf{e}_i \in \mathbb{R}^d$ denotes the $i$th elementary basis vector, with 1 at the $i$th coordinate and 0s elsewhere. For any off-the-shelf FO optimizers, *e.g.*, stochastic gradient descent (SGD), if we replace the FO gradient estimate with the ZO gradient estimate, then we obtain the ZO counterpart of a FO solver, *e.g.*, ZO-SGD (Ghadimi & Lan, 2013).

**Warm-up: A direct application of ZO optimization.** A straightforward method to achieve the DS-based black-box defense is to solve problem (2) using ZO optimization directly. However, it will give rise to the difficulty of *ZO optimization in high dimensions*. Specifically, DS requires to calculate the gradient of the defensive operation (1). With the aid of ZO gradient estimation, we obtain

$$\nabla_{\boldsymbol{\theta}}\mathcal{R}(f(\mathbf{x})) = \frac{dD_{\boldsymbol{\theta}}(\mathbf{x})}{d\boldsymbol{\theta}}\frac{df(\mathbf{z})}{d\mathbf{z}}\mid_{\mathbf{z}=D_{\boldsymbol{\theta}}(\mathbf{x})} \approx \frac{dD_{\boldsymbol{\theta}}(\mathbf{x})}{d\boldsymbol{\theta}}\hat{\nabla}_{\mathbf{z}}f(\mathbf{z})\mid_{\mathbf{z}=D_{\boldsymbol{\theta}}(\mathbf{x})}, \tag{5}$$

where with an abuse of notation, let $d$ denote the dimension of $\mathbf{x}$ (yielding $D_{\boldsymbol{\theta}}(\mathbf{x}) \in \mathbb{R}^d$ and $\mathbf{z} \in \mathbb{R}^d$) and $d_{\boldsymbol{\theta}}$ denote the dimension of $\boldsymbol{\theta}$, $\frac{dD_{\boldsymbol{\theta}}(\mathbf{x})}{d\boldsymbol{\theta}} \in \mathbb{R}^{d_{\boldsymbol{\theta}}\times d}$ is the Jacobian matrix of the vector-valued function $D_{\boldsymbol{\theta}}(\mathbf{x})$, and $\hat{\nabla}_{\mathbf{z}}f(\mathbf{z})$ denotes the ZO gradient estimate of $f$, following (3) or (4). Since the dimension of an input is typically large for image classification (*e.g.*, $d = 3072$ for a CIFAR-10 image), it imposes *two challenges*: (a) The variance of RGE (3) will be ultra-large if the query complexity stays low, *i.e.*, a small query number $q$ is used; And (b) the variance-least CGE (4) becomes impracticable due to the need of ultra-high querying cost (*i.e.*, $q = d$). Indeed, Table 1 shows that the direct application of (5) into the existing FO-DS solver (Salman et al., 2020), which we call ZO-DS, yields over 25% degradation in

| Method | Certified robustness (%) ($\ell_2$ radius: $\epsilon = 0.5$) | Standard accuracy (%) |
|---|---|---|
| FO-DS | 30.22 | 71.80 |
| ZO-DS (RGE, $q = 192$) | 5.06 ($\downarrow$ 25.16) | 44.81 ($\downarrow$ 26.99) |

Table 1: Performance comparison between FO-DS (Salman et al., 2020) and its direct ZO implementation ZO-DS on (CIFAR-10, ResNet-110).

both standard accuracy and certified robustness evaluated at input perturbations with $\ell_2$ norm less than $128/255$, where pixels of an input image are normalized to $[0, 1]$. We refer readers to Sec. 5 for more details.

**ZO autoencoder-based DS (ZO-AE-DS): A scalable solution to black-box defense.** The difficulty of ZO optimization in high dimensions prevents us from developing an effective DS-oriented provable defense for black-box ML models. To tackle such problem, we introduce an Autoencoder (AE) to connect the front-end denoiser $D_{\boldsymbol{\theta}}$ with the back-end black-box predictive model $f$ so that ZO optimization can be conducted in a (low-dimension) feature embedding

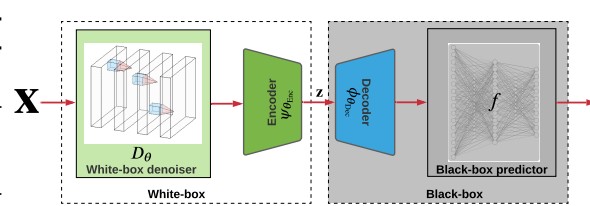

Figure 3: Model architecture for ZO-AE-DS.

space. To be concrete, let $\psi_{\boldsymbol{\theta}_{\mathrm{Dec}}} \circ \phi_{\boldsymbol{\theta}_{\mathrm{Enc}}}$ denote AE consisting of the encoder (Enc) $\psi_{\boldsymbol{\theta}_{\mathrm{Enc}}}$ and the

decoder (Dec) $\phi_{\boldsymbol{\theta}_{\text{Dec}}}$, where $\circ$ denotes the function composition operation. Plugging $\phi_{\boldsymbol{\theta}_{\text{AE}}}$ between the denoiser $D_{\boldsymbol{\theta}}$ and the black-box predictor $f$, we then extend the defensive operation (1) to the following (see Fig. 3 for illustration):

$$\mathcal{R}_{\text{new}}(f(\mathbf{x})) := \underbrace{f\left(\phi_{\boldsymbol{\theta}_{\text{Dec}}}(\mathbf{z})\right)}_{\text{new black box}}, \qquad \mathbf{z} = \underbrace{\psi_{\boldsymbol{\theta}_{\text{Enc}}}\left(D_{\boldsymbol{\theta}}(\mathbf{x})\right)}_{\text{new white box}}, \tag{6}$$

where $\mathbf{z} \in \mathbb{R}^{d_z}$ denotes the low-dimension feature embedding with $d_z < d$. In (6), we integrate the decoder $\psi_{\boldsymbol{\theta}_{\text{Dec}}}$ with the black-box predictor $f$ to construct a *new* black-box model $f'(\mathbf{z}) := f(\psi_{\boldsymbol{\theta}_{\text{Dec}}}(\mathbf{z}))$, which enables us to derive a ZO gradient estimate of *reduced dimension*:

$$\nabla_{\boldsymbol{\theta}} \mathcal{R}_{\text{new}}(f(\mathbf{x})) \approx \frac{d\phi_{\boldsymbol{\theta}_{\text{Enc}}}(D_{\boldsymbol{\theta}}(\mathbf{x}))}{d\boldsymbol{\theta}} \hat{\nabla}_{\mathbf{z}} f'(\mathbf{z}) \big|_{\mathbf{z}=\phi_{\boldsymbol{\theta}_{\text{Enc}}}(D_{\boldsymbol{\theta}}(\mathbf{x}))} . \tag{7}$$

Assisted by AE, RGE of $\hat{\nabla}_{\mathbf{z}} f'$ has a reduced variance from $O(\frac{d}{q})$ to $O(\frac{d_z}{q})$. Meanwhile, the least-variance CGE (4) also becomes feasible by setting the query number as $q = d_z$.

Note that the eventual ZO estimate (7) is a function of the $d_{\boldsymbol{\theta}} \times d_z$ Jacobian matrix $\nabla_{\boldsymbol{\theta}}[\phi_{\boldsymbol{\theta}_{\text{Enc}}}(D_{\boldsymbol{\theta}}(\mathbf{x}))]$. For ease of storing and computing the Jacobian matrix, we derive the following computationally-light alternative of (7) (see derivation in Appendix A):

$$\nabla_{\boldsymbol{\theta}} \ell_{\text{Stab}}(\boldsymbol{\theta}) \approx \nabla_{\boldsymbol{\theta}}[\mathbf{a}^{\top} \phi_{\boldsymbol{\theta}_{\text{Enc}}}(D_{\boldsymbol{\theta}}(\mathbf{x} + \boldsymbol{\delta}))], \quad \mathbf{a} = \hat{\nabla}_{\mathbf{z}} \ell_{\text{CE}}(f'(\mathbf{z}), f(\mathbf{x})) \big|_{\mathbf{z}=\phi_{\boldsymbol{\theta}_{\text{Enc}}}(D_{\boldsymbol{\theta}}(\mathbf{x}+\boldsymbol{\delta}))}, \tag{8}$$

where recall that $f'(\mathbf{z}) = f(\psi_{\boldsymbol{\theta}_{\text{Dec}}}(\mathbf{z}))$, and $\hat{\nabla}$ denotes the ZO gradient estimate given by (3) or (4). The computation advantage of (8) is that the derivative operation $\nabla_{\boldsymbol{\theta}}$ can be applied to a scalar-valued inner product built upon a pre-calculated ZO gradient estimate $\mathbf{a}$.

**Training ZO-AE-DS.** Recall from Fig. 3 that the proposed defensive system involves three components: denoiser $D_{\boldsymbol{\theta}}$, AE $\psi_{\boldsymbol{\theta}_{\text{Dec}}} \circ \phi_{\boldsymbol{\theta}_{\text{Enc}}}$, and pre-defined black-box predictor $f$. Thus, the parameters to be optimized include $\boldsymbol{\theta}$, $\boldsymbol{\theta}_{\text{Dec}}$ and $\boldsymbol{\theta}_{\text{Enc}}$. To train ZO-AE-DS, we adopt a two-stage training protocol. ① *White-box pre-training on AE*: At the first stage, we pre-train the AE model by calling a standard FO optimizer (*e.g.*, Adam) to minimize the reconstruction loss $\mathbb{E}_{\mathbf{x}} \|\phi_{\boldsymbol{\theta}_{\text{Dec}}}(\psi_{\boldsymbol{\theta}_{\text{Enc}}}(\mathbf{x})) - \mathbf{x}\|_2^2$. The resulting AE will be used as the initialization of the second-stage training. We remark that the denoising model $D_{\boldsymbol{\theta}}$ can also be pre-trained. However, such a pre-training could hamper optimization, *i.e.*, making the second-stage training over $\boldsymbol{\theta}$ easily trapped at a poor local optima. ② *End-to-end training*: At the second stage, we keep the pre-trained decoder $\phi_{\boldsymbol{\theta}_{\text{Dec}}}$ intact and merge it into the black-box system as shown in Fig. 3. We then optimize $\boldsymbol{\theta}$ and $\boldsymbol{\theta}_{\text{Enc}}$ by minimizing the DS-based training loss (2), where the denoiser $D_{\boldsymbol{\theta}}$ and the defensive operation $\mathcal{R}$ are replaced by $\psi_{\boldsymbol{\theta}_{\text{Enc}}} \circ D_{\boldsymbol{\theta}}$ and $\mathcal{R}_{\text{new}}$ (6), respectively. In (2), minimization over the stability loss $\ell_{\text{Stab}}(\boldsymbol{\theta})$ calls the ZO estimate of $\nabla_{\boldsymbol{\theta}} \ell_{\text{Stab}}(\boldsymbol{\theta})$, given by (7). In Appendix C.2, different training schemes are discussed.

## 5 EXPERIMENTS

In this section, we demonstrate the effectiveness of our proposal through extensive experiments. We will show that the proposed ZO-AE-DS outperforms a series of baselines when robustifying black-box neural networks for secure image classification and image reconstruction.

### 5.1 EXPERIMENT SETUP

**Datasets and model architectures.** In the task of image classification, we focus on CIFAR-10 and STL-10 datasets. In Appendix C.3, we demonstrate the effectiveness of ZO-AE-DS on the high-dimension ImageNet images. In the task of image reconstruction, we consider the MNIST dataset. To build ZO-AE-DS and its variants and baselines, we specify the prepended denoiser $D_{\boldsymbol{\theta}}$ as DnCNN (Zhang et al., 2017). We then implement task-specific AE for different datasets. Superficially, the dimension of encoded feature embedding, namely, $d_z$ in (6), is set as 192, 576 and 192 for CIFAR-10, STL-10 and MNIST, respectively. The architectures of AE are configured following (Mao et al., 2016), and ablation study on the choice of AE is shown in Appendix C.1. To specify the black-box image classification model, we choose ResNet-110 for CIFAR-10 following (Salman et al., 2020), and ResNet-18 for STL-10. It is worth noting that STL-10 contains 500 labeled $96 \times 96$ training images, and the pre-trained ResNet-18 achieves 76.6% test accuracy that matches to state-of-the-art performance. For image reconstruction, we adopt a reconstruction network consisting of convolution, deconvolution and ReLU layers, following (Raj et al., 2020).

**Baselines.** We will consider two *variants* of our proposed ZO-AE-DS: *i) ZO-AE-DS using RGE* (3), *ii) ZO-AE-DS using CGE* (4). In addition, we will compare ZO-AE-DS with *i) FO-AE-DS*, *i.e.*, the first-order implementation of ZO-AE-DS, *ii) FO-DS*, which developed in (Salman et al., 2020), *iii) RS*-based certified training, proposed in (Cohen et al., 2019), and *iv) ZO-DS*, *i.e.*, the ZO implementation of FO-DS using RGE. Note that CGE is not applicable to ZO-DS due to the obstacle of high dimensions. To our best knowledge, ZO-DS is the only query-based black-box defense baseline that can be directly compared with ZO-AE-DS.

**Training setup.** We build the training pipeline of the proposed ZO-AE-DS following 'Training ZO-AE-DS' in Sec. 4. To optimize the denoising model $D_{\boldsymbol{\theta}}$, we will cover two training schemes: training from scratch, and pre-training & fine-tuning. In the scenario of training from scratch, we use Adam optimizer with learning rate $10^{-3}$ to train the model for 200 epochs and then use SGD optimizer with learning rate $10^{-3}$ drop by a factor of 10 at every 200 epoch, where the total number of epochs is 600. As will be evident later, training from scratch over $D_{\boldsymbol{\theta}}$ leads to better performance of ZO-AE-DS. In the scenario of pre-training & fine-tuning, we use Adam optimizer to pre-train the denoiser $D_{\boldsymbol{\theta}}$ with the MSE loss $\ell_{\mathrm{Denoise}}$ in (2) for 90 epochs and fine-tune the denoiser with $\ell_{\mathrm{Stab}}$ for 200 epochs with learning rate $10^{-5}$ drop by a factor of 10 every 40 epochs. When implementing the baseline FO-DS, we use the best training setup provided by (Salman et al., 2020). When implementing ZO-DS, we reduce the initial learning rate to $10^{-4}$ for training from scratch and $10^{-6}$ for pre-training & fine-tuning to stabilize the convergence of ZO optimization. Furthermore, we set the smoothing parameter $\mu = 0.005$ for RGE and CGE. And to achieve a smooth predictor, we set the Gaussian smoothing noise as $\boldsymbol{\delta} \in \mathcal{N}(\mathbf{0}, \sigma^2\mathbf{I})$ with $\sigma^2 = 0.25$. With the help of matrix operations and the parallel computing power of the GPU, we optimize the training time to an acceptable range. The averaged one-epoch training time on a single Nvidia RTX A6000 GPU is about $\sim$ 1min and $\sim$ 29min for FO-DS and our proposed ZO method, ZO-AE-DS (CGE, $q = 192$), on the CIFAR-10 dataset.

**Evaluation metrics.** In the task of robust image classification, the performance will be evaluated at standard test accuracy (SA) and certified accuracy (CA). Here CA is a provable robust guarantee of the Gaussian smoothing version of a predictive model. Let us take ZO-AE-DS as an example, the resulting smooth image classifier is given by $f_{\mathrm{smooth}}(\mathbf{x}) := \arg\max_c \mathbb{P}_{\boldsymbol{\delta}\in\mathcal{N}(\mathbf{0},\sigma^2\mathbf{I})}[\mathcal{R}_{\mathrm{new}}(f(\mathbf{x} + \boldsymbol{\delta})) = c]$, where $\mathcal{R}_{\mathrm{new}}$ is given by (6). Further, a *certified radius* of $\ell_2$-norm perturbation ball with respect to an input example can be calculated following the RS approach provided in (Cohen et al., 2019). As a result, CA at a given $\ell_2$-radius $r$ is the percentage of the correctly classified data points whose certified radii are larger than $r$. Note that if $r = 0$, then CA reduces to SA.

| | FO | | | ZO-DS | | | ZO-AE-DS (Ours) | | | |
|---|---|---|---|---|---|---|---|---|---|---|
| $\ell_2$-radius $r$ | RS | FO-DS | FO-AE-DS | $q = 20$ (RGE) | $q = 100$ (RGE) | $q = 192$ (RGE) | $q = 20$ (RGE) | $q = 100$ (RGE) | $q = 192$ (RGE) | $q = 192$ (CGE) |
| 0.00 (SA) | **76.44** | 71.80 | 75.97 | 19.50 | 41.38 | 44.81 | 42.72 | 58.61 | 63.13 | **72.23** |
| 0.25 | **60.64** | 51.74 | 59.12 | 3.89 | 18.05 | 19.16 | 29.57 | 40.96 | 45.69 | **54.87** |
| 0.50 | **41.19** | 30.22 | 38.50 | 0.60 | 4.78 | 5.06 | 17.85 | 24.28 | 27.84 | **35.50** |
| 0.75 | **21.11** | 11.87 | 18.18 | 0.03 | 0.32 | 0.30 | 8.52 | 9.45 | 10.89 | **16.37** |

Table 2: SA (standard accuracy, %) and CA (certified accuracy, %) versus different values of $\ell_2$-radius $r$. Note that SA corresponds to the case of $r = 0$. In both FO and ZO blocks, the best accuracies for each $\ell_2$-radius are highlighted in **bold**.

## 5.2 EXPERIMENT RESULTS ON IMAGE CLASSIFICATION

**Performance on CIFAR-10.** In Table 2, we present certified accuracies of ZO-AE-DS and its variants/baselines versus different $\ell_2$-radii in the setup of (CIFAR-10, ResNet-110). Towards a comprehensive comparison, different RGE-based variants of ZO-AE-DS and ZO-DS are demonstrated using the query number $q \in \{20, 100, 192\}$. First, the comparison between ZO-AE-DS and ZO-DS shows that our proposal significantly outperforms ZO-DS ranging from the low query number $q = 20$ to the high query number $q = 192$ when RGE is applied. Second, we observe that the use of CGE yields the best CA and SA (corresponding to $r = 0$). The application of CGE is benefited from AE, which reduces the dimension from $d = 32 \times 32 \times 3$ to $d_z = 192$. In particular, CGE-based ZO-AE-DS improves the case studied in Table 1 from $5.06\%$ to $35.5\%$ at the $\ell_2$-radius $r = 0.5$. Third, although FO-AE-DS yields CA improvement over FO-DS in the white-box context, the improvement achieved by ZO-AE-DS (vs. ZO-DS) for black-box defense is much more significant. This implies

that the performance of black-box defense relies on a proper solution (namely, ZO-AE-DS) to tackle the challenge of ZO optimization in high dimensions. Fourth, RS outperforms the ZO methods. This is not surprising since RS is a known white-box certifiably robust training approach. In Appendix B, we demonstrate the consistent effectiveness of ZO-AE-DS under different denoiser and classifiers.

**Performance on STL-10.** In Table 3, we evaluate the performance of ZO-AE-DS for STL-10 image classification. For comparison, we also represent the performance of FO-DS, FO-AE-DS, and ZO-DS. Similar to Table 2, the improvement brought by our proposal over ZO-DS is evident, with at least 10% SA/CA improvement across different $\ell_2$-radii.

When comparing ZO-AE-DS with FO-DS, we observe that ours introduces a 7% degradation in SA (at $r = 0$). This is different from CIFAR-10 classification. There might be two reasons for the degradation of SA in STL-10. First, the size of a STL-10 image is $9\times$ larger than a CIFAR-10 image.

| | | | STL-10 | |
|---|---|---|---|---|
| $\ell_2$-radius $r$ | FO-DS | FO-AE-DS | ZO-DS (RGE, $q = 576$) | ZO-AE-DS (CGE, $q = d_z = 576$) |
| 0.00 (SA) | 53.36 | 54.26 | 38.60 | 45.67 |
| 0.25 | 35.83 | 43.99 | 21.50 | 35.78 |
| 0.50 | 21.61 | 34.85 | 9.58 | 26.70 |
| 0.75 | 9.86 | 25.56 | 3.29 | 17.91 |

Table 3: CA (certified accuracy, %) vs. different $\ell_2$-radii for image classification on STL-10.

Thus, the over-reduced feature dimension could hamper SA. In this example, we set $d_z = 576$, which is only $3\times$ larger than $d_z = 192$ used for CIFAR-10 classification. Second, the variance of ZO gradient estimates has a larger effect on the performance of STL-10 than that of CIFAR-10, since the former only contains 500 labeled images, leading to a challenging training task. Despite the degradation of SA, ZO-AE-DS outperforms FO-DS in CA, especially when facing a large $\ell_2$-radius. This is consistent with Table 2. The rationale is that AE can be regarded as an extra smoothing operation for the image classifier, and thus improves certified robustness over FO-DS, even if the latter is designed in a white-box setup. If we compare ZO-AE-DS with FO-AE-DS, then the FO approach leads to the best performance due to the high-accuracy of gradient estimates.

**Advantage of AE on ZO optimization.** Extended from Table 2, Fig. 4 presents the complete CA curve of non-AE-based and AE-based methods vs. the value of $\ell_2$-radius in the example of (CIFAR-10, ResNet-110). As we can see, ZO-AE-DS using RGE with the smallest query number $q = 20$ has outperformed ZO-DS using RGE with the largest query number $q = 192$. This shows the vital role of AE on ZO optimization. Meanwhile, consistent with Table 2 and Table 3, the best model achieved by ZO-AE-DS using CGE could be even better than the FO baseline FO-DS since AE could play a similar role on the smoothing operation. Furthermore, as the query number $q$ increases, the improvement of ZO-AE-DS grows, towards the performance of FO-AE-DS.

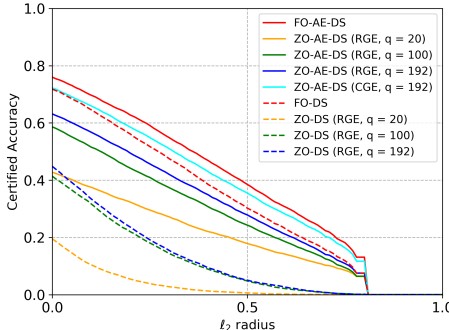

Figure 4: Comparison between non-AE-based and AE-based methods in CA vs. different $\ell_2$-radii. Dashed lines: Models obtained by non-AE-based methods; Solid lines: Models obtained by AE-based methods.

**Effect of training scheme on ZO-AE-DS.** In Table 4, we present the impact of training scheme (over the denoiser $D_{\theta}$) on the CA performance of ZO-AE-DS versus different $\ell_2$-radii. Two training schemes, training from scratch and pre-training + fine-tuning, are considered. As we can see, training from scratch for $D_{\theta}$ leads to the better performance of ZO-AE-DS than pre-training + fine-tuning.

| | ZO-AE-DS (CGE, $q = 192$) | |
|---|---|---|
| $\ell_2$-radius $r$ | Training from scratch | Pre-training + fine-tuning |
| 0.00 | 72.23 | 59.74 |
| 0.25 | 54.87 | 42.61 |
| 0.50 | 35.50 | 26.26 |
| 0.75 | 16.37 | 11.13 |

Table 4: ZO-AE-DS using different denoiser training schemes under (CIFAR-10, ResNet-110).

This is because the application of pre-training to $D_{\theta}$ could make optimization easily get trapped at a local optima. We list other ablation studies in Appendix C.

### 5.3 EXPERIMENT RESULTS ON IMAGE RECONSTRUCTION.

In what follows, we apply the proposed ZO-AE-DS to robustifying a black-box image reconstruction network. The goal of image reconstruction is to recover the original image from a noisy measurement.

Following (Antun et al., 2020; Raj et al., 2020), we generate the noisy measurement following a linear observation model $\mathbf{y} = \mathbf{A}\mathbf{x}$, where $\mathbf{A}$ is a sub-sampling matrix (*e.g.*, Gaussian sampling), and $\mathbf{x}$ is an original image. A pre-trained image reconstruction network (Raj et al., 2020) then takes $\mathbf{A}^\top \mathbf{y}$ as the input to recover $\mathbf{x}$. To evaluate the reconstruction performance, we adopt two metrics (Antun et al., 2020), the root mean squared error (RMSE) and structural similarity (SSIM). SSIM is a supplementary metric to RMSE, since it gives an accuracy indicator when evaluating the similarity between the true image and its estimate at fine-level regions. The vulnerability of image reconstruction networks to adversarial attacks, *e.g.*, PGD attacks (Madry et al., 2018), has been shown in (Antun et al., 2020; Raj et al., 2020; Wolf, 2019).

When the image reconstructor is given as a black-box model, spurred by above, Table 5 presents the performance of image reconstruction using various training methods against adversarial attacks with different perturbation strengths. As we can see, compared to the normally trained image reconstructor (*i.e.*, 'Standard' in Table 5), all robustification methods lead to degraded standard image reconstruction performance in the non-adversarial context (*i.e.*, $\|\boldsymbol{\delta}\|_2 = 0$). But the worst performance is provided by ZO-DS. When the perturbation strength increases, the model achieved by standard training becomes over-sensitive to adversarial perturbations, yielding the highest RMSE and the lowest SSIM. Furthermore, we observe that the proposed black-box defense ZO-AE-DS yields very competitive and even better performance with respect to FO defenses. In Fig. 5, we provide visualizations of the reconstructed images using different approaches at the presence of reconstruction-evasion PGD attacks. For example, the comparison between Fig. 5-(f) and (b)/(d) clearly shows the robustness gained by ZO-AE-DS.

| Image reconstruction on MNIST | | | | | | | | | | |
|---|---|---|---|---|---|---|---|---|---|---|
| Method | $\|\boldsymbol{\delta}\|_2 = 0$ | | $\|\boldsymbol{\delta}\|_2 = 1$ | | $\|\boldsymbol{\delta}\|_2 = 2$ | | $\|\boldsymbol{\delta}\|_2 = 3$ | | $\|\boldsymbol{\delta}\|_2 = 4$ | |
| | RMSE | SSIM | RMSE | SSIM | RMSE | SSIM | RMSE | SSIM | RMSE | SSIM |
| Standard | 0.112 | 0.888 | 0.346 | 0.417 | 0.493 | 0.157 | 0.561 | 0.057 | 0.596 | 0.014 |
| FO-DS | 0.143 | 0.781 | 0.168 | 0.703 | 0.221 | 0.544 | 0.278 | 0.417 | 0.331 | 0.337 |
| ZO-DS | 0.197 | 0.521 | 0.217 | 0.474 | 0.262 | 0.373 | 0.313 | 0.284 | 00.356 | 0.225 |
| FO-AE-DS | 0.139 | 0.792 | 0.162 | 0.717 | 0.215 | 0.554 | 0.274 | 0.421 | 0.329 | 0.341 |
| ZO-AE-DS | 0.141 | 0.79 | 0.164 | 0.718 | 0.217 | 0.551 | 0.277 | 0.42 | 0.33 | 0.339 |

Table 5: Performance of image reconstruction using different methods at various attack scenarios. Here 'standard' refers to the original image reconstructor without making any robustification. Four robustification methods are presented including FO-DS, ZO-DS (RGE, $q = 192$), FO-AE-DS, and ZO-AE-DS (CGE, $q = 192$). The performance metrics RMSE and SSIM are measured by adversarial example $(\mathbf{x} + \boldsymbol{\delta})$, generated by 40-step $\ell_2$ PGD attacks under different values of $\ell_2$ perturbation norm $\|\boldsymbol{\delta}\|_2$.

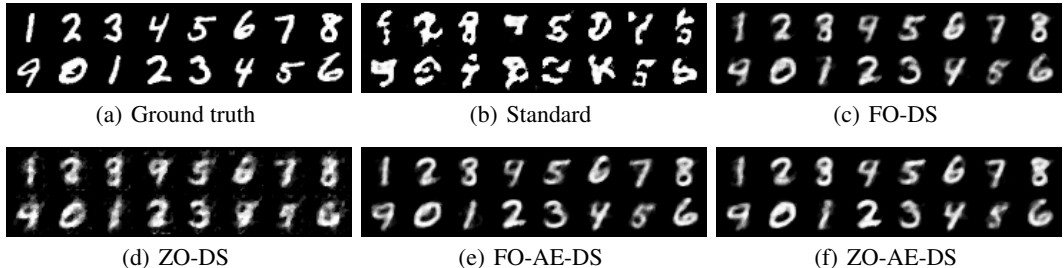

(a) Ground truth     (b) Standard     (c) FO-DS

(d) ZO-DS     (e) FO-AE-DS     (f) ZO-AE-DS

Figure 5: Visualization for Image Reconstruction under $\ell_2$ PGD attack (Step $= 40$, $\epsilon = 1.0$ ). Original: base reconstruction network. ZO-DS: RGE with $q = 192$. ZO-AE-DS: CGE with $q = 192$

## 6 CONCLUSION

In this paper, we study the problem of black-box defense, aiming to secure black-box models against adversarial attacks using only input-output model queries. The proposed black-box learning paradigm is new to adversarial defense, but is also challenging to tackle because of the black-box optimization nature. To solve this problem, we integrate denoised smoothing (DS) with ZO (zeroth-order) optimization to build a feasible black-box defense framework. However, we find that the direct application of ZO optimization makes the defense ineffective and difficult to scale. We then propose ZO-AE-DS, which leverages autoencoder (AE) to bridge the gap between FO and ZO optimization. We show that ZO-AE-DS reduces the variance of ZO gradient estimates and improves the defense and optimization performance in a significant manner. Lastly, we evaluate the superiority of our proposal to a series of baselines in both image classification and image reconstruction tasks.

ACKNOWLEDGMENT

Yimeng Zhang, Yuguang Yao, Jinghan Jia, and Sijia Liu are supported by the DARPA RED program.

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

## A  DERIVATION OF (8)

First, based on (2) and (6), the stability loss corresponding to ZO-AE-DS is given by

$$\ell_{\text{Stab}}(\boldsymbol{\theta}) = \ell_{\text{CE}}\left(f'(\mathbf{z}), f(\mathbf{x})\right) := g(\mathbf{z}), \text{ where } f'(\mathbf{z}) = f\left(\phi_{\boldsymbol{\theta}_{\text{Dec}}}(\mathbf{z})\right), \ \mathbf{z} = \psi_{\boldsymbol{\theta}_{\text{Enc}}}\left(D_{\boldsymbol{\theta}}(\mathbf{x} + \boldsymbol{\delta})\right). \tag{9}$$

We then take the derivative of $\ell_{\text{Stab}}(\boldsymbol{\theta})$ w.r.t. $\boldsymbol{\theta}$. This yields

$$\nabla_{\boldsymbol{\theta}} \ell_{\text{Stab}}(\boldsymbol{\theta}) = \frac{d\mathbf{z}}{d\boldsymbol{\theta}} \frac{dg(\mathbf{z})}{d\mathbf{z}} \big|_{\mathbf{z} = \psi_{\boldsymbol{\theta}_{\text{Enc}}}(D_{\boldsymbol{\theta}}(\mathbf{x}+\boldsymbol{\delta}))}, \tag{10}$$

where $\frac{d\mathbf{z}}{d\boldsymbol{\theta}} \in \mathbb{R}^{d_{\theta} \times d}$ and $\frac{dg(\mathbf{z})}{d\mathbf{z}} \in \mathbb{R}^d$.

Since $g(\mathbf{z})$ involves the black-box function $f$, we first compute its ZO gradient estimate following (3) or (4) and obtain

$$\frac{dg(\mathbf{z})}{d\mathbf{z}} \big|_{\mathbf{z} = \psi_{\boldsymbol{\theta}_{\text{Enc}}}(D_{\boldsymbol{\theta}}(\mathbf{x}+\boldsymbol{\delta}))} \approx \hat{\nabla}_{\mathbf{z}} g(\mathbf{z}) \big|_{\mathbf{z} = \psi_{\boldsymbol{\theta}_{\text{Enc}}}(D_{\boldsymbol{\theta}}(\mathbf{x}+\boldsymbol{\delta}))} := \mathbf{a}. \tag{11}$$

Substituting the above into (10), we obtain

$$\nabla_{\boldsymbol{\theta}} \ell_{\text{Stab}}(\boldsymbol{\theta}) = \frac{d\mathbf{z}}{d\boldsymbol{\theta}} \mathbf{a} = \begin{bmatrix} \frac{d\mathbf{a}^{\top}\mathbf{z}}{d\theta_1} \\ \frac{d\mathbf{a}^{\top}\mathbf{z}}{d\theta_2} \\ \vdots \\ \frac{d\mathbf{a}^{\top}\mathbf{z}}{d\theta_{d_{\theta}}} \end{bmatrix} = \frac{d\mathbf{a}^{\top}\mathbf{z}}{d\boldsymbol{\theta}} = \nabla_{\boldsymbol{\theta}}[\mathbf{a}^{\top}\phi_{\boldsymbol{\theta}_{\text{Enc}}}(D_{\boldsymbol{\theta}}(\mathbf{x}+\boldsymbol{\delta}))], \tag{12}$$

where the last equality holds based on (9). This completes the derivation.

## B  COMBINATION OF DIFFERENT DENOISERS AND CLASSIFIERS

Table A1 presents the certified accuracies of our proposal using different denoiser models (Wide-DnCnn vs. DnCnn) and image classifier (Vgg-16).

| $\ell_2$-radius $r$ | DnCnn & VGG-16 | | | Wide-DnCnn & VGG-16 | | |
|---|---|---|---|---|---|---|
| | FO-DS | FO-AE-DS | ZO-AE-DS (CGE, $q = d_z = 192$) | FO-DS | FO-AE-DS | ZO-AE-DS (CGE, $q = d_z = 192$) |
| 0.00 (SA) | 71.37 | 73.75 | 71.92 | 66.57 | 75.14 | 72.97 |
| 0.25 | 51.37 | 54.74 | 54.33 | 50.1 | 57.45 | 54.92 |
| 0.50 | 30.21 | 34.6 | 34.39 | 31.52 | 37.59 | 34.2 |
| 0.75 | 11.72 | 15.45 | 15.36 | 13.94 | 17.64 | 15.7 |

Table A1: CA (certified accuracy, %) vs. different $\ell_2$-radii for different combinations of denoisers and classifier.

## C  ADDITIONAL EXPERIMENTS AND ABLATION STUDIES

In what follows, we will show the ablation study on the choice of AE architectures in Appendix C.1. Afterwards, we will show the performance of FO-AE-DS versus different training schemes in Appendix C.2. Finally, we will show the performance of our proposal on the high-dimension ImageNet images in Appendix C.3.

### C.1  THE PERFORMANCE OF FO-AE-DS WITH DIFFERENT AUTOENCODERS.

Table. A2 presents the certified accuracy performance of FO-AE-DS with different autoencoders (AE). As we can see, if AE-96 is used (namely, the encoded dimension is half of AE-192 used in the paper), then we observe a slight performance drop. This is a promising result as we can further reduce the query complexity by choosing a different autoencoder since the use of CGE has to be matched with the encoded dimension.

| $\ell_2$-radius $r$ | AE-96 | AE-192 |
|---|---|---|
| 0.00 (SA) | 75.57 | **75.97** |
| 0.25 | 58.07 | **59.12** |
| 0.50 | 37.09 | **38.50** |
| 0.75 | 17.05 | **18.18** |

Table A2: CA (certified accuracy, %) vs. different $\ell_2$-radii for FO-AE-DS with different AutoEncoders.

## C.2 THE PERFORMANCE OF FO-AE-DS WITH DIFFERENT TRAINING SCHEMES.

Table. A3 presents the certified accuracy of FO-AE-DS (first-order implementation of ZO-AE-DS) with different training schemes. Training both denoiser and encoder is the default setting. As we can see, only training the denoiser would bring performance degradation, and training both denoiser and AE does boost the performance. It is worth noting that FO-AE-DS with "train the denoiser and AE" training scheme can be regarded as the FO-DS treating the combination of the original denoiser and the same AE used in FO-AE-DS as a new denoiser, which cannot be implemented for ZO-AE-DS since the decoder of ZO-AE-DS is merged into the black-box classifier and its parameters cannot be updated. Furthermore, the key of the introduced AE is to reduce the variable dimension for Zeroth-Order (ZO) gradient estimation.

| $\ell_2$-radius $r$ | FO-DS | FO-AE-DS (only train the denoiser) | FO-AE-DS (train the denoiser and encoder) | FO-AE-DS (train the denoiser and the AE) |
|---|---|---|---|---|
| 0.00 (SA) | 71.80 | 73.34 | 75.97 | 75.76 |
| 0.25 | 51.74 | 55.61 | 59.12 | 58.14 |
| 0.50 | 30.22 | 35.68 | 38.50 | 38.88 |
| 0.75 | 11.87 | 15.92 | 18.18 | 18.48 |

Table A3: CA (certified accuracy, %) vs. different $\ell_2$-radii for FO-AE-DS with different training schemes.

## C.3 THE PERFORMANCE OF ZO-AE-DS ON IMAGENET IMAGES.

To evaluate the performance of ZO-AE-DS on the Restricted ImageNet (R-ImageNet) dataset, a 10-class subset of ImageNet with 38472 images for training and 1500 images for testing, similar to (Tsipras et al., 2019). Due to our limited computing resources, we are not able to scale up our experiment to the full ImageNet dataset, but the purpose of evaluating on high-dimension images remains the same. In the implementation of ZO-AE-DS, we choose an AE with an aggressive compression (130:1), which is to compress the original $3 \times 224 \times 224$ images into the $1152 \times 1 \times 1$ feature dimension. We compare the certified accuracy (CA) performance of our proposed ZO-AE-DS (using CGE) with the black-box baseline ZO-DS, and the white-box baselines FO-DS and FO-AE-DS. Results are summarized in the following table.

As we can see, (1) when considering the black-box classifier, the proposed ZO-AE-DS still significantly outperforms the direct ZO implementation of DS. This shows the importance of variance reduction of query-based gradient estimates. (2) Since ZO-AE-DS and FO-AE-DS used an aggressive AE structure, the performance drops compared to FO-DS. (3) the use of high-resolution images would make the black-box defense much more challenging. However, ZO-AE-DS is still a principled black-box defense method that can achieve reasonable performance.

| $\ell_2$-radius $r$ | FO-DS | FO-AE-DS | ZO-AE-DS (RGE, q =1152 and encoder) | ZO-AE-DS (CGE, q=1152) |
|---|---|---|---|---|
| 0.00 (SA) | 89.33 | 71.07 | 26.93 | 63.60 |
| 0.25 | 81.67 | 63.40 | 18.40 | 52.80 |
| 0.50 | 68.87 | 53.60 | 11.67 | 43.13 |
| 0.75 | 49.80 | 42.87 | 5.53 | 32.73 |

Table A4: CA (certified accuracy, %) vs. different $\ell_2$-radii for FO-AE-DS on ImageNet Images.

