# OpenReview forum: "How to Robustify Black-Box ML Models? A Zeroth-Order Optimization Perspective"
_ICLR.cc/2022/Conference — ICLR 2022 Spotlight_

### Official Review · Reviewer_RU4Z · 2021-10-26

**Correctness:** 3
**Technical Novelty And Significance:** 3
**Empirical Novelty And Significance:** 3
**Recommendation:** 8
**Confidence:** 3

**Main Review:**

The contribution of this work may be relatively incremental. This is because the black-box defense setting has been considered in [1]. The differences between this work and [1] are: 1) query-based black-box defense is studied in this work 2) the authors introduce an AutoEncoder to reduce the variance of gradient estimation. Moreover, introducing AutoEncoder to promote gradient estimation has been studied in [2].

Maybe, above comments are somewhat harsh. In fact, I have felt that. Thus, if the authors can highlight the contribution of this work, especially compared with [1], I am willing to raise my rating.

I guess one contribution of this work may be that the authors have shown that query-based black-box defense is simple yet effective and improving the efficiency of gradient estimation is a good strategy for the query-based black-box defense setting.

[1] Denoised Smoothing: A Provable Defense for Pretrained Classifiers. NeurIPS2020.
[2] AutoZOOM: Autoencoder-based Zeroth Order Optimization Method for Attacking Black-box Neural Networks. AAAI2019.


**Summary Of The Paper:**

This paper explores an exciting direction, i.e., black-box defense, where defense methods cannot access parameters of the target model when endowing the target model with certified adversarial robustness. However, the black-box defense setting considered in this paper has been studied in [1].

**Summary Of The Review:**

I like this work, but I do hope the authors can further highlight the contribution, especially when comparing this work with [1].

---

> ### Author Response · Authors · 2021-11-19
> **Response to Reviewer RU4Z**
>
> Thank you for liking our paper! We are encouraged that you find our paper interesting. It is our pleasure to answer the following questions.
>
> **Q1: Highlight the contribution of this work, especially compared with [Denoised Smoothing](https://arxiv.org/pdf/2003.01908.pdf) [1].**
>
> **A1:** Thank you very much for the comment. In our original submission, the difference between Denoised Smoothing and our proposed method was described in the third paragraph of the Introduction. Following your suggestion, we will provide more details below.
>
> Our paper aims to design the **restriction-least** black-box defense using just input-output model queries. By contrast, denoised smoothing tackled the black-box defense problem by leveraging an ensemble of surrogate models. Yet, this still requires to have access to the information on the victim model type and its function. In practice, those conditions could be difficult to achieve. For example, in the domain of medical ML, if the domain knowledge related to medicine or healthcare is lacking (Qayyum et al., 2020; Finlayson et al., 2019), then it will be difficult to determine a proper surrogate model of the original medical ML system. To the best of our knowledge, there has been no prior work in the design of query-based defense. And it is non-trivial to have a principled ZO designing framework and demonstrate its effectiveness not only for image classification but also for image reconstruction.
>
> References:
>
> Adnan Qayyum, Junaid Qadir, Muhammad Bilal, and Ala Al-Fuqaha. Secure and robust machine learning for healthcare: A survey. IEEE Reviews in Biomedical Engineering, 14:156–180, 2020.
>
> Samuel G Finlayson, John D Bowers, Joichi Ito, Jonathan L Zittrain, Andrew L Beam, and Isaac S Kohane. Adversarial attacks on medical machine learning. Science, 363(6433):1287–1289, 2019.
>
>
> **Q2: In [Autozoom](https://arxiv.org/abs/1805.11770) [2], introducing AutoEncoder to promote gradient estimation has been studied.**
>
> **A2:** It is a great comment. Although Autozoom does employ the AutoEncoder to reduce dimension for Zeroth-Order gradient estimation, our proposed method ZO-AE-DS is different from it, as expanded below.
>
>
> 1) First of all, Autozoom focuses on **black-box adversarial attack** generation, but we tackle the problem of **black-box defense**, which is much more challenging as the model parameters need to be trained using ZO optimization.
> 2) Secondly, Autozoom employs the white-box decoder to map the generated low-dimension perturbation back to the original input dimension. However, we introduced the ZO-AE structure, where the decoder needs to be merged into the black-box system so as to tackle the high-dimension challenge of ZO model training. And then, we need to train both the denoiser and the encoder to achieve the black-box defense strategy.
>
>
> [1]Cohen, Jeremy, Elan Rosenfeld, and Zico Kolter. "Certified adversarial robustness via randomized smoothing." International Conference on Machine Learning. PMLR, 2019.
>
> [2]Tu, Chun-Chen, et al. "Autozoom: Autoencoder-based zeroth-order optimization method for attacking black-box neural networks." Proceedings of the AAAI Conference on Artificial Intelligence. Vol. 33. No. 01. 2019.

---

> > ### Comment · Reviewer_RU4Z · 2021-11-22
> > **My concerns have been addressed**
> >
> > The authors' excellent rebuttal addresses all my concerns and I have improved my score to 8.

---

### Official Review · Reviewer_ATPZ · 2021-11-02

**Correctness:** 4
**Technical Novelty And Significance:** 4
**Empirical Novelty And Significance:** 3
**Recommendation:** 8
**Confidence:** 4

**Main Review:**

Pros:
1.	The concept of black-box defense is novel and interesting, and this paper is the first work to tackle the problem of query-based black-box defense.
2.	The author shows the idea of how to solve the black-box defense step by step. The paper ingeniously applies the zero-order optimization method to the DS method and leverages autoencoder to bridge the gap between first-order and zero-order optimization.
3.  The paper is clearly written and easy to follow.


Cons:
1.	The experiments are conducted on CIFAR-10 and STL-10. Also, It can be observed that the larger the picture, the worse the defense effect. Is this method still effective on ImageNet? If ZO-AE-DS cannot be applied to ImageNet, then the method is of little significance. After all, black-box defense is designed to solve the problem in real-world scenes.
2.	The references are all before 2020. The authors should add more recent works.
3.	The baselines are not strong enough. The experiments are all compared with DS and its variants.


**Summary Of The Paper:**

The authors formulate the problem of black-box defense and propose a novel black-box defense approach called the Zero Order AutoEncoder-based Denoised Smoothing (ZO-AE-DS). Black-box defense corresponds to situations in which the defense model information cannot be obtained due to privacy protection in real scenarios. ZO-AE-DS introduces zero-order optimization on the structure of denoised smoothing (DS) to estimate the gradient and uses an Autoencoder (AE) to connect the denoiser with the model so that zero-order optimization can be conducted in a (low-dimension) feature embedding space.

**Summary Of The Review:**

Overall, this paper formulates the problem of black-box defense and provides a feasible black-box defense framework. It is clearly written and easy to follow.The updated experiments are comprehensive and convincing.

[Update]: Though the method still can not really handle ImageNet, the authors' rebuttal addresses most of my concerns. Also, the setting of black-box defenses is novel and should be encouraged. Hence, I am increasing my score to 8.

---

> ### Author Response · Authors · 2021-11-19
> **Response to Reviewer ATPZ**
>
> Thank you very much for the insightful and positive feedback! We are encouraged that you find our proposed ZO-AE-DS to be a novel method and query-based black-box defense to be a valuable task. In what follows, we will provide a detailed response to your questions.
>
> **Q1: Can ZO-AE-DS be applied to high-dimension ImageNet images?**
>
> **A1:** To address your concern, we added a **new experiment** to evaluate the performance of ZO-AE-DS on the Restricted ImageNet (R-ImageNet) dataset, a 10-class subset of ImageNet with 38472 images for training and 1500 images for testing, similar to [1].
> Due to our limited computing resources, we are not able to scale up our experiment to the full ImageNet dataset, but the purpose of evaluating on high-dimension images remains the same. In the implementation of ZO-AE-DS,  we choose an AutoEnocoder with an aggressive compression (130:1), which is to compress the original $3 \times 224 \times 224$ images into the $1152 \times 1 \times 1$ feature dimension.
> We compare the certified accuracy (CA) performance of our proposed ZO-AE-DS (using CGE) with the black-box baseline ZO-DS, and the white-box baselines FO-DS and FO-AE-DS. Results are summarized in the following table.
>
>
> |  $\ell_2$-radius $r$  | FO-DS| FO-AE-DS| ZO-DS (RGE, q=1152)| ZO-AE-DS  (CGE, q=1152)|
> |  :----:  | :---:   |:---:   |:---:   |:---:   |
> |0.00 | 89.33 |71.07  |26.93  |  63.60 |
> | 0.25 |81.67 | 63.40 |18.40 | 52.80  |
> | 0.50 |68.87 | 53.60 | 11.67 | 43.13  |
> | 0.75 |49.80 | 42.87| 5.53| 32.73 |
>
> As we can see, (1) when considering the black-box classifier, the proposed ZO-AE-DS still significantly outperforms the direct ZO implementation of DS. This shows the importance of variance-reduced ZO gradient estimates. (2) Since ZO-AE-DS and FO-AE-DS used an aggressive AE structure, the performance drops compared to FO-DS. (3) We agree with the reviewer that the use of high-resolution images will make the black-box defense much more challenging. However, ZO-AE-DS is still a principled black-box defense method that can achieve reasonable performance.
>
>
> **Q2:  The authors should add more recent works.**
>
> **A2:** Thanks for pointing this out. We would add more recent work. However, we kindly remark that we have tried our best to list the most relevant references.
>
> **Q3: The baselines are not strong enough. The experiments are all compared with DS and its variants.**
>
> **A3:** According to your advice, we add the Randomized Smoothing (RS)-based certified training [2] as a stronger white-box baseline. In the following table, we compare the certified accuracy (CA) performance of our proposed black-box defense ZO-AE-DS with that of FO-AE-DS and RS, the currently strongest first-order baseline. As we can see, ZO-AE-DS indeed provides the comparable CA even against RS. We highlight that ZO-AE-DS is built upon just using model queries.
>
> |  $\ell_2$-radius $r$  | RS  | FO-AE-DS| ZO-AE-DS  (CGE, q=192)|
> |  :----:   |:---:   |:---:   |:---:   |
> |0.00 (SA)| 76.44  | 75.97   |  72.23  |
> | 0.25 | 60.64 |  59.12  | 54.87   |
> | 0.50 |  41.19   |  38.50  | 35.50   |
> | 0.75 |  21.11   |  18.18  |  16.37  |
>
> [1]Tsipras, Dimitris, Shibani Santurkar, Logan Engstrom, Alexander Turner, and Aleksander Madry. "Robustness may be at odds with accuracy." arXiv preprint arXiv:1805.12152 (2018).
>
> [2]Cohen, Jeremy, Elan Rosenfeld, and Zico Kolter. "Certified adversarial robustness via randomized smoothing." International Conference on Machine Learning. PMLR, 2019.

---

> ### Author Response · Authors · 2021-11-29
> **Kindly check in the last day**
>
> Dear Reviewer ATPZ,
>
> Thank you very much for spending time on reviewing our paper. Since the discussion phase will end very soon. We kindly check if our  response has addressed your previous questions/comments. If you have other questions/comments, please feel free to let us know. We will try our best to reply to you before the discussion deadline.
>
> Thank you very much,
>
> Authors

---

### Official Review · Reviewer_ULvk · 2021-11-02

**Correctness:** 3
**Technical Novelty And Significance:** 4
**Empirical Novelty And Significance:** 4
**Recommendation:** 8
**Confidence:** 5

**Main Review:**

Strength:
The the proposed ZO-AE-DS paradigm is novel to me. The idea of black-box defense is very interesting, which is actually a very important real-world application scenario. Compared to the competing method FO-DS, the authors eliminate the need for white-box model by applying zero-order optimization, and also use an autoencoder to avoid dimension issues for calculation. Then they demonstrate it with experiments that AE can also improve robustness. The paper is overall easy to follow, and clearly written. Experiments are thorough and well organized.

Weakness:
The performance of ZO-AE-DS (zero-order) does not outperform FO-AE-DS (first-order) since the latter needs a surrogate model whereas ZO only requires access to model’s queries. The question is that it is not too hard to obtain a surrogate model if we have queries access. This paper can be further strengthened if benefit of avoiding surrogate models are articulated.

Typo: “a recently-developed A2-type approach (Salman et al., 2020),” should be “R2-type”


**Summary Of The Paper:**

This paper proposes a novel approach to robustify black-box models to address the problem of black-box defense, which arises due to the concerns of privacy. The approach proposed by the authors essentially incorporates zero-order optimization on a prepended denoising smoothing(DS) module, with an autoencoder connecting the DS and model, termed as “ZO-AE-DS”, which the authors claim that it can robustify models using only model queries.

**Summary Of The Review:**

Question: Both ZO-AE-DS and FO-AE-DS outperforms their counterparts without AE module. Considering the denoising nature of AE, the DS+AE can be viewed as a type of stacked DS. Have the authors tried other combinations of stacked DS? Is it possible to also increase the performance like inserting AE to FO-DS? Overall I think the proposed zero-order optimization for black-box defense is a very promising application scenario. Although the method seems doesn’t outperform FO version with surrogates, I think it has broader applicability owning to its minimum assumptions.

---

> ### Author Response · Authors · 2021-11-19
> **Response to Reviewer ULvk**
>
> Thank you very much for your recognition of our work. Please see our detailed response to your insightful comments below.
>
> **Q1:  The performance of ZO-AE-DS (zeroth-order) does not outperform FO-AE-DS (first-order) since the latter needs a surrogate model, whereas ZO only requires access to the model's queries. It is not too hard to obtain a surrogate model if we have query access. Why do we need to avoid the use of the surrogate model for black-box defense?**
>
> **A1:** This is a very valuable comment and provides a good future research direction.
>
> 1) We agree with the reviewer that if a black-box model can be reverse engineered or well approximated using a surrogate model, then yes, we can apply the first-order (FO) approach. However, finding a high-quality surrogate model could be very difficult in practice. To achieve such a model estimate, an extremely large number of queries are needed since query complexity is proportional to the dimension of optimization variables (namely, model parameters).
> 2) In privacy-demanded applications (such as AI for healthcare), the victim model is private and could be difficult to be reverse engineered using just model queries. Moreover, it is very difficult to find a surrogate model when the knowledge on the prediction task is lacking.
>
>
>
> **Q2: Typo: a recently-developed $\mathcal A_2$-type approach.**
>
>
> **A2:** Thank you very much for pointing out this typo. It should be $\mathcal R_2$.
>
> **Q3: How about the performance of Denoised Smoothing using a different denoiser?**
>
> **A3:** It is a good comment. We did try to employ a different denoiser for our proposed method.
> 1) Firstly, we have tried to replace DnCnn (the default denoiser choice for our experiments) with  Wide-DnCnn, which yet brings performance degradation. The details are shown in the Appendix of our paper.
> 2) Secondly, following your suggestion, we treat a combination of the original denoiser and the same AutoEncoder used in FO-AE-DS as a **new denoiser** in the context of Denoised Smoothing (FO-DS). We then evaluate the certified accuracy performance of FO-DS with the new denoiser in the table below. It does boost the robustness performance of FO-DS.
>
>
> |  $\ell_2$-radius $r$  |FO-AE-DS (original denoiser)|FO-DS (original denoiser)|FO-DS (new denoiser)|
> |:---:|:---:|:---:|:---:|
> |0.00| 75.97 |71.80|75.75|
> |0.25| 59.12 |51.74|58.14|
> |0.50|38.50 |30.22|38.88|
> |0.75|18.18 |11.87|18.48|

---

> > ### Comment · Reviewer_ULvk · 2021-11-21
> > **The response is thorough and clear**
> >
> > The authors’ response is thorough and clear. Regarding A1, in the final version it would be nice if the authors can provide an analysis on why ZO approach need less queries than surrogate, and some detailed explanation (or cite other works) on scenarios that surrogate approaches are infeasible.

---

> > > ### Author Response · Authors · 2021-11-22
> > > **Response to Reviewer ULvk's Revision Suggestion**
> > >
> > > Thank you very much for your prompt response. Following your suggestion, we add more supporting references on our argument for the case of lacking domain knowledge and the intensive querying cost of model inversion.
> > >
> > > The modified paragraph together with references is repeated below:
> > >
> > > Yet, this still requires to have access to the information on the victim model type and its function. In practice, those conditions could be difficult to achieve. For example, if the domain knowledge related to medicine or healthcare is lacking (Qayyum et al., 2020; Finlayson et al., 2019), then it will be difficult to determine a proper surrogate model of a medical ML system. Even if a black-box model estimate can be obtained using the model inversion technique (Kumar & Levine, 2019), a significantly large number of model queries are needed even just for tackling an MNIST-level prediction task (Oh et al., 2019).
> > >
> > >
> > > References:
> > >
> > > Adnan Qayyum, Junaid Qadir, Muhammad Bilal, and Ala Al-Fuqaha. Secure and robust machine learning for healthcare: A survey. IEEE Reviews in Biomedical Engineering, 14:156–180, 2020.
> > >
> > > Samuel G Finlayson, John D Bowers, Joichi Ito, Jonathan L Zittrain, Andrew L Beam, and Isaac S Kohane. Adversarial attacks on medical machine learning. Science, 363(6433):1287–1289, 2019.
> > >
> > > Aviral Kumar and Sergey Levine. Model inversion networks for model-based optimization. arXiv preprint arXiv:1912.13464, 2019.
> > >
> > > Seong Joon Oh, Bernt Schiele, and Mario Fritz. Towards reverse-engineering black-box neural networks. In Explainable AI: Interpreting, Explaining and Visualizing Deep Learning, pp. 121–144. Springer, 2019.

---

### Official Review · Reviewer_Vo8C · 2021-11-03

**Correctness:** 3
**Technical Novelty And Significance:** 3
**Empirical Novelty And Significance:** 3
**Recommendation:** 8
**Confidence:** 4

**Main Review:**

Strengths:
- They show comparison with the baselines over a sufficiently wide range of experiments.
- The argument for the use of the Autoencoder is intuitive and clearly works. The algorithm itself is simple, building on tools that have been known to work.
- The algorithm itself is truly black-box in that it does not require knowledge of the model that is being robustified.

Concerns/Comments:
- Though it is fair that the algorithm is only compared against other black-box algorithms, it would be useful to show the certified accuracy(CA) of a whitebox algorithm like (Cohen et al., Madry et al.) as a comparison. This would illustrate the gap between the two regimes.
- While the autoencoder reduces the variance of the gradient estimates, the fact that $2q$ forward passes need to be made for every gradient computation, should indicate that turning this method into a ZO-method increases the computational burden. I did not see this discussed in the paper. A comparison would be useful.
- I believe there is a typo on page 4 where $\mathcal{A}_2$ is used when it should be $\mathcal{R}_2$. Either choice of nomenclature is fine as long as it is consistent.
- Is there any justification for choosing $q=192$ in the experiments on Cifar10?
- The authors mention that pre-training $D_{\theta}$ causes it to get stuck at a local optima. However, this doesn't seem to happen to $\theta_{ENC}$. I was also curious to see if any experiments were conducted where the autoencoder was fixed after pre-training and only $\theta$ was trained. Does this perform poorly?
- Is my understanding correct in that the accuracies reported in Table 2 are over different sets of data? It seems like the algorithm outputs a certified radius for each point and we compute the CA over only points which have a certified radius greater than $r$. I find this confusing because Salman et al. report a SA for every $r$ while in this setting SA only makes sense for $r=0$. Can you clarify?
- In Table 2, RGE and CGE for $q=192$ are quite different. However, if the AE converges correctly then the random directions in RGE should all be linearly independent and provide an equivalent estimate to the standard basis directions. Is there any intuition for this difference?

**Summary Of The Paper:**

This work provides an algorithm to ensure robust training of an ML model with just black-box knowledge of it i.e., input and output access. The algorithm relies on using Denoised Smoothing with zeroth-order optimization where the gradients are estimated using random perturbations (finite-differencing). They avoid the computational burden and high variance of these estimates by first training an auto-encoder to reduce the inputs to a low-dimensional subspace. They show over different architectures and multiple datasets (Cifar10, STL10, image reconstruction over MNIST) that the proposed algorithm performs better than the baseline which is the Denoised Smoothing algorithm by Salman et al.

**Summary Of The Review:**

Overall, I find the algorithm to make sense intuitively and the experiments are quite thorough. The results are superior to the current baselines and therefore I am satisfied with this paper.

~However, I have a few concerns and clarifications. If they are resolved, I would recommend the paper to be accepted.~
[Update]: I am satisfied with the author's response. I am increasing my score.

---

> ### Author Response · Authors · 2021-11-19
> **Response to Reviewer Vo8C (Part II)**
>
> **Q4: Why is $q=192$ chosen for the experiments on the CIFAR-10 dataset?**
>
> **A4:** This is because if we choose the Coordinate-wise Gradient Estimate (CGE), we must make the quantity of query number $q$ the same as the dimension of the feature embedding (namely, the output of encoder in ZO-AE-DS) of which the gradient is estimated. The output dimension of our chosen encoder for the CIFAR-10 dataset is $192$, so we choose $q=192$ for experiments on the CIFAR-10 dataset. To delve into this further, we added a new experiment (see table below) to compare the certified accuracy performance of FO-AE-DS under different AutoEncoders (AE-192 vs. AE-96). As we can see, if AE-96 is used (namely, the encoded dimension is half of AE-192 used in the paper), then we observe a slight performance drop. This is a promising result as we can further reduce the query complexity by choosing a different autoencoder since the use of CGE has to be matched with the encoded dimension.
>
> |  $\ell_2$-radius $r$  | AE-96 | AE-192 |
> |  :----:  | :---:   |:---:   |
> |0.00  | 75.57  |**75.97** |
> | 0.25   |  58.07  | **59.12** |
> | 0.50 | 37.09 |**38.50**  |
> | 0.75  | 17.05  | **18.18** |
>
>
>
> **Q5: How about the performance of FO-AE-DS if only the denoiser is trained and the AutoEncoder is fixed?**
>
> **A5:** It can still achieve reasonable performance. However, its performance is worse than training both denoiser and encoder. The certified accuracy evaluation details are shown in the table below.
>
> |  $\ell_2$-radius $r$  |FO-AE-DS (Only train denoiser)|FO-AE-DS (train denoiser and encoder)|
> |:---:|:---:|:---:|
> |0.00 |73.34 | **75.97** |
> |0.25|55.61 | **59.12** |
> |0.50|35.68 | **38.50** |
> |0.75| 15.92 | **18.18** |
>
>
> **Q6: Is my understanding correct in that the accuracies reported in Table 2 are over different sets of data? And clarification on Standard Accuracy (SA) and Certified Accuracy (CA).**
>
> **A6:** Table 2 is only corresponding to the CIFAR-10 dataset. And yes, the algorithm would output a certified radius, and only if this certified radius is greater than the given $\ell_2$-radius, then the classifier at this $\ell_2$ perturbation radius is called ``"certified"``.
>
> "Why does [1] report a SA for every $\ell_2$-radius $r$?" This is a good question. We apologize that we did not make this clear.
>
> The work of Denoised Smoothing reported different standard accuracies at different $\ell_2$-radius $r$ because they used different noise levels $\sigma$ to smooth the model for evaluation and select the best one at different $\ell_2$-radius $r$. In other words, their results at different $\ell_2$-radius $r$ may not correspond to the same smooth classifier. By contrast, in our experiments, we intended to report and compare our results under the consistent smooth classifier. This makes the standard accuracy independent of the perturbation radius. Please feel free to refer to Appendix B of [1] for more details.
>
> **Q7: Why is there a performance difference between Random Gradient Estimate (RGE) and Coordinate-wise Gradient Estimate (CGE) when the same query number $q$ is chosen for them?**
>
> **A7:** Even if the same $q$ is used in RGE and CGE, however, their variance is still different. The former has the larger variance due to the use of stochastic random direction vectors; see theoretical justification in [2]. This is why ZO-AE-DS using CGE can achieve better performance than ZO-AE-DS using RGE even if they choose the same query number $q$.
>
>
> [1]Salman, Hadi, et al. "Denoised smoothing: A provable defense for pretrained classifiers." arXiv preprint arXiv:2003.01908 (2020).
>
> [2]Liu, Sijia, et al. "Zeroth-order stochastic variance reduction for nonconvex optimization." arXiv preprint arXiv:1805.10367 (2018).

---

> > ### Comment · Reviewer_Vo8C · 2021-11-27
> > **Response to Rebuttal (Very Satisfied)**
> >
> > I see. The experiments with a reduced dimension are again quite surprising. It would be interesting to see how low the intrinsic dimension of the feature embedding really is.
> >
> > Again, I am very satisfied with this response.

---

> > > ### Author Response · Authors · 2021-11-27
> > > **Thank you!**
> > >
> > > Dear Reviewer Vo8C,
> > >
> > > Thank you very much for the follow-up suggestion. We will try to find the lowest intrinsic dimension by testing the performance of our approach under a more diverse setup of feature embeddings. And it is our great pleasure to see that you are very satisfied with our response.
> > >
> > > Thanks,

---

> ### Author Response · Authors · 2021-11-19
> **Response to Reviewer Vo8C (Part I)**
>
> Thanks for the insightful and positive feedback! In what follows, we address your comments point by point.
>
> **Q1: Comparison with the white-box training baseline like (Cohen et al., Madry et al.) in Certified accuracy (CA).**
>
> **A1:** As suggested by your comment, we have evaluated the certified accuracy (CA) of the ResNet-110 model trained using randomized smoothing (RS) [1] and using vanilla adversarial training (AT) [2]. Note that the RS-trained ResNet-110 was provided in [1], and the AT variant of ResNet-110 was not available in the literature. Thus, we trained the latter by ourselves, leading to the 89.5%  standard accuracy and the 76.2% empirical robust accuracy against $L_2$ PGD attack ($step =20$ and $\epsilon=64/255$). Once these ResNet-110 models are achieved, CA is then evaluated under their smoothed classifiers at the testing time, as shown in the following table, where AT and RS refer to the smooth version of RS-based ResNet-110 and AT-based ResNet-110, respectively.
>
> |  $\ell_2$-radius $r$  | AT| RS | FO-DS  | FO-AE-DS| ZO-AE-DS  (CGE, q=192)|
> |  :----:  | :---:   |:---:   |:---:   |:---:   |:---:   |
> |0.00| 11.07 | 76.44  |  71.80| 75.97   |  72.23  |
> | 0.25 | 8.66 | 60.64  | 51.74 |  59.12  | 54.87   |
> | 0.50 | 5.39 | 41.19  | 30.22  |  38.50  | 35.50   |
> | 0.75 | 1.15 | 21.11  | 11.87 |  18.18  |  16.37  |
>
> The above table shows that the RS approach outperforms the ZO methods and the AT-based empirical defense. This is not surprising since RS is a known white-box certifiably robust training approach. And the AT-based model is not trained using random noise and thus is not smoothing-aware. Thus, its test-time CA performance (based on its smooth counterpart) becomes the worst. Compared to RS, our proposed ZO-AE-DS can also provide comparable CA performance even without having access to the knowledge of the classifier.
>
> **Q2: Computation cost comparison between FO-DS and ZO-AE-DS.**
>
> **A2:** As the reviewer pointed out, our approach does require $2q$ times forward passes, which is unavoidable for our zeroth-order query-based approach to reduce the ZO gradient estimate variance. However, the good news is that we can optimize the training time to an acceptable range using matrix operations and the parallel computing power of the GPU. Finally, the averaged one-epoch training time on a single Nvidia RTX A6000 GPU is about ~$1$min and ~$29$min for FO-DS and our proposed ZO method, ZO-AE-DS(CGE, $q=192$) on the CIFAR-10 dataset, respectively. The increasing factor of the computation time is less than the increasing factor of the query number $q$.
>
> **Q3: Typo on page 4.**
>
> **A3:** Yes. It should be **$\mathcal R_2$**. Thank you very much for the careful reading!
>
> [1]Cohen, Jeremy, Elan Rosenfeld, and Zico Kolter. "Certified adversarial robustness via randomized smoothing." International Conference on Machine Learning. PMLR, 2019.
>
> [2]Madry, Aleksander, et al. "Towards deep learning models resistant to adversarial attacks." arXiv preprint arXiv:1706.06083 (2017).

---

> > ### Comment · Reviewer_Vo8C · 2021-11-27
> > **Response to Rebuttal (Very Satisfied)**
> >
> > Thank you for running so many more experiments. I understand that it is an extremely demanding ask during the rebuttal phase, so I really appreciate it.
> > The fact that ZO-AE-DS comes so close to RS is very impressive. I would recommend including these results in the paper (with the clarification that AT is at a disadvantage here since it is unaware of smoothing. Or perhaps leave out AT).
> >
> > The computational tradeoff is not ideal, but it is to be expected when we ask for comparable results without first order information. All of this makes perfect sense. I am satisfied as long as these clarifications are included (in some form) in the final version.

---

> > > ### Author Response · Authors · 2021-11-27
> > > **Thank you!**
> > >
> > > Dear Reviewer Vo8C,
> > >
> > > Thank you very much for the follow-up and for raising the score. We will be sure to include the suggested discussion in the final version.
> > >
> > > Thanks,

---

### Author Response · Authors · 2021-11-19
**General Response**

Thank you very much for the very insightful comments. The paper has been revised based on reviewers’ comments.

Before making the detailed response, we would like to summarize our newly conducted experiment inspired by reviewers' suggestions.

**Summary of newly conducted experiments.**
1) [Reviewer Vo8C:](https://openreview.net/forum?id=W9G_ImpHlQd&noteId=XXmBfknc8nJ) Following your comments, we added the [comparison](https://openreview.net/forum?id=W9G_ImpHlQd&noteId=VrANlcWMMhq) with randomized smoothing (RS) [1] and the vanilla adversarial training (AT). In addition, we evaluated the performance of FO-AE-DS [using different AutoEncoders](https://openreview.net/forum?id=W9G_ImpHlQd&noteId=o3LP2p0QajU). Furthermore, we obtained a [FO-AE-DS by only training the denoiser](https://openreview.net/forum?id=W9G_ImpHlQd&noteId=o3LP2p0QajU) and evaluate its performance.
2) [Reviewer ULvk:](https://openreview.net/forum?id=W9G_ImpHlQd&noteId=yrdsOmXB8K_) Following your comments, we added the comparison with a new FO-DS model, in which the denoiser consists of the original denoiser (DnCnn) and the AutoEncoder used for ZO-AE-DS.
3) [Reviewer ATPZ:](https://openreview.net/forum?id=W9G_ImpHlQd&noteId=SgMswovWjNv) Following your comments, we evaluated our proposed method ZO-AE-DS on the high-dimension ImageNet images. In addition, we added the comparison with the RS baseline [1].

[1] Cohen, Jeremy, Elan Rosenfeld, and Zico Kolter. "Certified adversarial robustness via randomized smoothing." International Conference on Machine Learning. PMLR, 2019.

---

### Decision · Program_Chairs · 2022-01-20

**Decision:**

Accept (Spotlight)

**Comment:**

All reviewers have converged to an unanimous rating of the paper, highlighting, in the paper or during the discussion, many strengths, including a compelling approach clearly relevant to applications and its solid range of experiments.

A clear accept, and I would encourage the authors to push in the final version the experiments and discussions following the threads with reviewers (in particular, Vo8C and ULvk).

Thanks also to authors and reviewers for a thorough discussion which helped to strengthen further the paper's content.

AC.